# In Vivo Reflectance Confocal Microscopy as a Response Monitoring Tool for Actinic Keratoses Undergoing Cryotherapy and Photodynamic Therapy

**DOI:** 10.3390/cancers13215488

**Published:** 2021-10-31

**Authors:** Clara Curiel-Lewandrowski, Caitlyn N. Myrdal, Kathylynn Saboda, Chengcheng Hu, Edith Arzberger, Giovanni Pellacani, Franz Josef Legat, Martina Ulrich, Petra Hochfellner, Margaret C. Oliviero, Paola Pasquali, Melissa Gill, Rainer Hofmann-Wellenhof

**Affiliations:** 1Division of Dermatology, The University of Arizona College of Medicine, Tucson, AZ 85724, USA; cmyrdal@arizona.edu; 2The University of Arizona Cancer Center, Tucson, AZ 85724, USA; ksaboda@uacc.arizona.edu; 3Department of Epidemiology and Biostatistics, Mel and Zuckerman College of Public Health, The University of Arizona, Tucson, AZ 85721, USA; hucc@arizona.edu; 4Department of Dermatology, Medical University of Graz, 8036 Graz, Austria; edith.arzberger@medunigraz.at (E.A.); franz.legat@medunigraz.at (F.J.L.); petra-hochfellner@gmx.at (P.H.); rainer.hofmann@medunigraz.at (R.H.-W.); 5Dermatology, Department of Clinical Internal, Anesthesiological and Cardiovascular Sciences, La Sapienza University of Rome, 00185 Rome, Italy; pellacani.giovanni@unimo.it; 6CMB Collegium Medicum Berlin GmbH/Dermatology Office, 10117 Berlin, Germany; ulrich@dermatologie-am-regierungsviertel.de; 7Skin and Cancer Associates, Fort Lauderdale, FL 33324, USA; maggie@admcorp.com; 8Pius Hospital of Valls, 43850 Tarragona, Spain; ppasquali@piushospital.cat; 9Faculty of Medicine and Health Sciences, University of Alcalá de Henares, 28801 Madrid, Spain; melgill@hotmail.com; 10Department of Pathology, SUNY Downstate Health Sciences University, Brooklyn, NY 11203, USA

**Keywords:** reflectance confocal microscopy, actinic keratosis, response monitoring, cryotherapy, photodynamic therapy, RCM

## Abstract

**Simple Summary:**

The assessment of actinic keratoses (AKs) in prevention and therapeutic trials, as well as clinical practice, could significantly benefit from the incorporation of non-invasive imaging technology. Such technology has the potential to enhance the objective evaluation of clinical and subclinical AKs with the added advantage of sequential monitoring. In vivo reflectance confocal microscopy (RCM) allows for the non-invasive imaging of AKs at a cellular level. We aimed to establish an in in vivo RCM protocol for AK response monitoring, ultimately leading to more reliable characterization of longitudinal responses and therapy optimization.

**Abstract:**

Reflectance confocal microscopy (RCM) presents a non-invasive method to image actinic keratosis (AK) at a cellular level. However, RCM criteria for AK response monitoring vary across studies and a universal, standardized approach is lacking. We aimed to identify reliable AK response criteria and to compare the clinical and RCM evaluation of responses across AK severity grades. Twenty patients were included and randomized to receive either cryotherapy (*n* = 10) or PDT (*n* = 10). Clinical assessment and RCM evaluation of 12 criteria were performed in AK lesions and photodamaged skin at baseline, 3 and 6 months. We identified the RCM criteria that reliably characterize AK at baseline and display significant reduction following treatment. Those with the highest baseline odds ratio (OR), good interobserver agreement, and most significant change over time were atypical honeycomb pattern (OR: 12.7, CI: 5.7–28.1), hyperkeratosis (OR: 13.6, CI: 5.3–34.9), stratum corneum disruption (OR: 7.8, CI: 3.5–17.3), and disarranged epidermal pattern (OR: 6.5, CI: 2.9–14.8). Clinical evaluation demonstrated a significant treatment response without relapse. However, in grade 2 AK, 10/12 RCM parameters increased from 3 to 6 months, which suggested early subclinical recurrence detection by RCM. Incorporating standardized RCM protocols for the assessment of AK may enable a more meaningful comparison across clinical trials, while allowing for the early detection of relapses and evaluation of biological responses to therapy over time.

## 1. Introduction

Actinic keratoses are common skin lesions with the risk to evolve into squamous cell carcinoma (SCC). It is currently not possible to predict which AK lesions will progress and various sources differ on the potential rate of progression. A landmark paper from Callen et al. approximated a 0.25–20% annual transformation rate for a single AK [1]. Cockerell estimated that the 10-year neoplastic transformation rate for the average person with 7 AKs is between 10–20% [2]. The diagnosis of AK is typically made via clinical evaluation and those that appear suspicious for SCC are biopsied for histopathologic confirmation.

Treatment options for AK are abundant and include photodynamic therapy (PDT), cryotherapy, 5-fluorouracil, diclofenac gel, imiquimod, and chemical peels, among others. Cryotherapy represents a lesion-directed therapy while PDT is a field-directed therapy that requires a photosensitizer, light, and oxygen. Photosensitizing agents include topical methyl aminolevulinate (MAL), 5-aminolevulinic acid (ALA), or BF-200 ALA, a nanoemulsion-based gel that optimizes epidermal penetration [3]. While light sources typically include red or blue light, daylight PDT has also been demonstrated to be effective [4,5].

To date, preventive and therapeutic approaches targeting AK have primarily relied on clinical and histological assessment to measure the effectiveness of the intervention used. Both approaches have significant limitations, including the low sensitivity and subjectivity of clinical assessment and the invasiveness associated with skin biopsy procedures, which hampers the possibility for the longitudinal monitoring of individual lesion response.

Reflectance confocal microscopy is a non-invasive method to image the skin at the cellular level using an in vivo approach and allows for a resolution similar to that of histology. RCM can be used to diagnose AK with 80% sensitivity and 98% specificity [6]. Furthermore, it has been suggested that RCM can detect early morphological changes and cellular atypia before an AK is clinically apparent [7,8,9]. Hence, due to its non-invasive nature, RCM is advantageous as subclinical AK can be detected and monitored over a period of time with relative ease, and little discomfort for the patient. Pellacani et al. demonstrated good interobserver correlation and high concordance between RCM and histological grading of keratinocyte atypia, highlighting that RCM can be as effective as a histopathologic examination [10].

Studies have also examined the use of RCM technology to follow the outcomes of therapeutic interventions in AK. In 2018, Ishioka et al. described a reduction in the RCM parameters of nucleated polygonal cells and isolated keratinocytes with 5% fluorouracil treatment [11]. Pasquali et al. monitored AK treatment with sequential cryotherapy and ingenol mebutate, identifying an atypical honeycomb pattern, round papillary vessels, parakeratosis, and stratum corneum detachment as reliable RCM parameters for response characterization [12]. In 2019, Benati et al. identified a significant improvement of keratinocyte disarray, parakeratosis, and crust following topical imiquimod treatment [13]. The above studies all concluded that RCM is effective for AK response monitoring; however, methods of evaluating responses varied considerably. Inconsistency of reliable RCM parameter identification is also seen in non-therapeutic AK characterization studies at baseline [14,15,16,17,18]. The establishment of a standard, universally accepted protocol for the evaluation of AK treatment responses by RCM could greatly equalize results between trials and add clinical benefit to surveillance.

The primary goal of our study was to contribute to this need by examining RCM criteria for AK and identifying those that display a significant response to treatment, good interobserver agreement, and high prevalence at baseline when compared to nearby photodamaged (PD) skin. Our secondary aims were to compare RCM and the clinical evaluation of responses, evaluate differences across AK grades, and assess cryotherapy and PDT treatments of AK as measured by RCM. Unique contributions to the current state of the field include the incorporation of nearby photodamaged (PD) skin as a reference for baseline assessment, comparison of RCM parameters across AK grades, and utilization of three experienced confocal evaluators to assess interobserver agreement.

## 2. Materials and Methods

### 2.1. Study Population

This single-center prospective study included patients recruited from the Dermatology Department at the Medical University of Graz, Austria. A total of 20 healthy patients aged 45–85 years with Fitzpatrick skin types I-II and a clinical diagnosis of two AKs (3–6 mm in diameter, grades 1–2) on the forehead, scalp, hands, or forearms were included. All participants signed an informed consent form, and the study was approved by the institutional review board and conducted in accordance with the Declaration of Helsinki principles. 

### 2.2. Study Design

Patients were randomly assigned to receive either treatment with cryotherapy (*n* = 10) or photodynamic therapy (PDT, *n* = 10). Three study skin sites were selected for evaluation: 1. Grade 1 AK (G1); 2. Grade 2 AK (G2); 3. Photodamaged skin (PD) at a separate location and in close proximity to the AKs. Clinical grading was completed through assessment of hyperkeratosis from 0 (absent) to 3 (3+), based on the Olsen et al. grading criteria [19]. Lesions were graded based on palpability: 0 = not palpable when compared to surrounding skin, 1 = slight palpability (the AK is felt better than seen), 2 = moderate palpability (the AK is seen and felt), and 3 = severe (the AK easily is visualized and felt). For more granular clinical scoring, measures of erythema and hyperkeratosis were scored with half-points ranging from 0 to 3 to characterize lesions more accurately. For instance, 0.5 = trace palpability (lesion cannot be seen visually and is minimally palpable) and 2.5 = moderate palpability but not considered severe. Measures of erythema ranged from 0 = none, 0.5 = minimal, 1 = mild, 2 = moderate, 3 = severe. Clinical scoring was performed by one investigator, CCL. Skin sites were photographed at each visit by standard protocol, which included regional photography followed by close-up and dermoscopy images. In addition, detailed measurements (3 per lesion) indicating the location of the study lesions were obtained. Individuals agreed to limit significant sun exposure to study areas at least 15 days prior to a scheduled imaging session and agreed to wear protective clothing when outdoors.

Study skin sites were evaluated clinically, by RCM, and by the standard imaging protocol at baseline, 3 and 6 months post-treatment. For both groups, the first visit consisted of baseline photographic documentation and RCM followed by treatment administration to AKs.

RCM imaging was performed using a confocal microscope (Vivascope 1500–Mavig) for which details of this technique have been described previously [20,21]. Four horizontally mapped mosaic images (VivaBlocks) were obtained at representative levels of the stratum corneum, mid epidermis, dermoepidermal junction and papillary dermis. Only areas without artifacts within the mosaics were scored. Two vertically mapped images (VivaStacks) were obtained per lesion, one each from the left upper and right lower quadrants of the field of view captured with RCM, with regular step intervals from the surface of the stratum corneum to the dermis. Stratum corneum thickness was calculated by counting the number of step intervals from the top of the stratum corneum to the top of the stratum granulosum. Stratum corneum thickness >20 µm on the face and >40 µm on other sites was considered hyperkeratosis [22].

### 2.3. Treatment Protocol

#### 2.3.1. Cryotherapy

AKs were treated using a Brymill’s Cry-Ac^®^ 500 cc hand-held device (Brymill Cryogenic Systems, Ellington, CT, USA). Liquid nitrogen was delivered through a C aperture (Model #102-C, 0.022in.). The surface temperature was measured using an IR (Infrared) thermometer and each lesion was frozen until −22 to −25 °C was obtained. Spraying was perpendicular to the skin and 5 cm from the skin surface. The depth of freezing front was measured using 22 MHz high-frequency ultrasound immediately after freezing to conform that the ice block had reached the dermoepidermal junction. 

#### 2.3.2. PDT

Topical methyl aminolevulinate (MAL) pre-light photosensitizer was administered to AK lesions after removal of the superficial crust. Three hours later, the area was irradiated with a red-light source (light dose 37 J/cm² from a medical lamp Aktilite^®^ CL128 (Galderma Laboritories, Lausanne, Switzerland), emitting heat-free, visible red light at a peak wavelength of 630 nm; irradiation time was 8 min) to activate the photosensitizer. The PD skin sites were not directly treated in either group.

### 2.4. Reflectance Confocal Microscopy Evaluation

RCM analysis included the scoring of 12 diagnostic criteria for AK based on previous work which examined features through RCM [8,16]. All parameters were scored as either “absent” or “present” except for presence of round blood vessels, polymorphous blood vessels, inflammatory infiltrate dermis (0 = absent, 1 = 1+, 2 = 2+, 3 = 3+), and atypical honeycomb pattern (0 = absent, 1 = mild, 2 = moderate, 3 = severe). To assess baseline interobserver variability, RCM readers (GP, CCL, RHW) were asked to score images from a subset of 10 study subjects at baseline through which a preliminary proportion of agreement was calculated for each criterion. After the consensus meetings, the drafting of a reference document incorporating consensus definitions, full subject enrollment, and a washout period of approximately 4 months, the RCM analysis was completed on all cases using the reference document. Interobserver proportion of agreement was calculated again. Study RCM image analysis was completed by three experienced RCM evaluators (cryotherapy group: GP, CCL/EA, MU; PDT group: GP, CCL/EA, MO). At the Graz study site, images were read together by CCL and EA.

### 2.5. Statistical Methods

Dichotomous RCM parameters were compared using logistic regression with GEE to adjust for intrasubject correlations. Ordinal RCM parameters were compared using ordinal logistic regression, with clustering within subjects to adjust for intrasubject correlations. Clinical features were scored on a semi-quantitative scale from 0–3. For comparisons of clinical features, Kruskal-Wallis tests were used. The percentage of agreement is presented with binomial 95% confidence intervals. Analyses were performed using Stata 16 (StataCorp, College Station, TX, USA).

## 3. Results

### 3.1. Patient Population

Twenty patients met the criteria and were randomized into the study. The mean age was 74 (SD 6.7) and 18 (90%) of the patients were male. Patients were recruited into the Dermatology Department at the Medical University of Graz, Austria; 19 had Fitzpatrick skin type II (95%), and one had Fitzpatrick skin type I (5%). A total of 40 AKs were evaluated and treated, one G1 AK and one G2 AK per individual. Fourteen AKs were located on the forehead (35%), thirteen elsewhere on the face (32.5%), seven on the scalp (17.5%), and six on the dorsal hand and forearm (15%).

### 3.2. Clinical Assessment of AKs

Clinically, AKs in both treatment groups demonstrated a significant decrease in the parameters of hyperkeratosis and erythema at 3 and 6 months (Table 1). In the cryotherapy group, treatment outcome at 6 months was evaluated clinically as a complete response in 8/10 G1 AKs (80%) and 8/10 G2 AKs (80%). A partial response at 6 months was seen in 2/10 G1 AKs (20%) and 2/10 G2 AKs (20%). In the PDT group, clinical evaluation at 6 months was scored as a complete response in 6/9 G1 AKs (67%), and 5/9 G2 AKs (56%). A partial response was seen in 3/9 G1 AKs (33%) and 4/9 G2 AKs (44%). Clinical grading scores were not available for one patient in the PDT group.

### 3.3. RCM Interobserver Agreement

Table 2 displays the RCM interobserver agreement. Prior to the start of this study and before the criteria standardization, the proportion of agreement between three experienced RCM readers ranged from 37 (CI: 24–52) to 87 (CI: 80–92). Standardization efforts were achieved through consensus meetings, in which criteria definitions and examples were discussed to implement consistency among readers. Post-standardization effort, the proportion of agreement between three experienced RCM readers, ranged from 61 (CI: 57–65) to 86 (CI: 83–88). Interobserver agreement improvements for stratum corneum disruption, round nucleated cells, disarranged epidermal pattern, and polymorphous blood vessels reached statistical significance following the standardization efforts. Parakeratosis and atypical honeycomb patterns approached significance. Interobserver agreement was excellent before and after standardization for atypical honeycomb pattern, intraepidermal dendritic cells, and dermal inflammatory cell infiltrate.

### 3.4. Baseline RCM Scoring

Twelve RCM criteria were evaluated per lesion. A comparison between AKs and adjacent PD skin yielded statistical significances in every parameter except for the presence of dendritic cells and round nucleated cells (Table 3). RCM criteria demonstrating the highest, most significant OR included hyperkeratosis (OR: 13.6, CI: 5.3–34.9), atypical honeycomb pattern (OR: 12.7, CI: 5.7–28.1), stratum corneum disruption (OR: 7.8, CI: 3.5–17.3), and disarranged epidermal pattern (OR: 6.5, CI: 2.9–14.8).

### 3.5. RCM Evaluation over Time

Table 4 shows RCM scores at 3 and 6 months following cryotherapy or PDT treatment.

#### 3.5.1. Stratum Corneum

Overall, parameters in the stratum corneum and epidermis exhibited the greatest response to treatment (Figure 1). In the stratum corneum of G1 AKs, 3/3 criteria in the cryotherapy group and 1/3 criteria in the PDT group displayed a significant reduction at 6 months compared to baseline. Hyperkeratosis was the only stratum corneum parameter that exhibited a statistically significant reduction at 6 months in G2 AKs, and this was seen in the cryotherapy treatment group. However, all (3/3) stratum corneum parameters in G2 AKs displayed a significant decrease at 3 months.

#### 3.5.2. Epidermis

In G1 AKs, 3/5 epidermal parameters in the cryotherapy group and 1/5 in the PDT group showed a significant reduction at 6 months compared to baseline. In G2 AKs, 2/5 epidermal parameters in both the cryotherapy and PDT groups displayed a significant reduction at 6 months.

#### 3.5.3. Dermis

In G1 AKs, both cryotherapy and PDT significantly reduced the intradermal inflammatory infiltrate at 6 months. In the PDT-treated G1 lesions, round blood vessels significantly increased at both 3 and 6 months. Measures of polymorphous vessels and solar elastosis did not significantly change over time.

RCM parameters with the greatest significance across groups included atypical honeycomb pattern, hyperkeratosis, disarranged epidermal pattern, and stratum corneum disruption (Appendix A). Of note, these four criteria also displayed the highest OR at baseline. Interestingly, various parameters in G2 AKs increased in prevalence from 3 to 6 months, likely representing early AK recurrence. Though these increases did not reach significance, this trend can be seen in 10/12 criteria in the PDT-treated G2 AKs, and 6/12 criteria in the cryotherapy-treated G2 AKs, with the most prominent increases observed in parakeratosis, stratum corneum disruption, and intraepidermal inflammatory cells.

### 3.6. Comparison between Treatment Groups

Overall, both treatments decreased AK severity as measured by RCM. Direct comparisons of the cryotherapy and PDT treatment group did not reach statistical significance in AK. Both the PDT and cryotherapy treatments were tolerated by all patients.

## 4. Discussion

RCM has the potential to standardize the therapeutic monitoring of actinic keratosis over time. However, the identification of relevant and reliable confocal parameters varies greatly across clinical trials, and a standardized protocol for RCM measurement of AK treatment response is needed. Although we observed significant reductions in many of the RCM criteria, our results suggest that atypical honeycomb patterns, hyperkeratosis, disarranged epidermal patterns, and stratum corneum disruptions are the most reliable criteria for assessing AK treatment response. These criteria displayed the highest baseline OR and exhibited significant responses to treatment across AK grades and treatment modalities. These results, taken together with recent publications by Benati et al., Ishioka et al., and Sousa et al., suggest that these criteria could be effectively implemented in future AK clinical studies using in vivo RCM as the primary monitoring tool for endpoint assessment [11,13,23]. It would be of significant benefit to simplify the RCM criteria used to date in order to maximize their adoption, and to incorporate evaluation of PD skin as an internal control.

Our identification of atypical honeycomb patterns as a reliable RCM parameter for AK is consistent with many reports highlighting the utility of keratinocyte disarray/atypical honeycomb patterns in AK response monitoring [11,12,13,23,24,25,26,27,28]. Hyperkeratosis [8,13,15,17,23,25,29] and stratum corneum disruption [8,11,12] have also been identified as significant RCM parameters, although many studies have not included these criteria in the RCM analysis of AK treatment responses. Interestingly, following its use in early clinical trials [6,16], future studies largely omitted disarranged epidermal pattern/epidermal disarray as an RCM criterion for AK, likely due to difficulty in standardization related to its varying definitions (ranging from cytological atypia, disordered keratinocyte maturation and/or architectural patterns) and challenges in its differentiation from similar epidermal parameters. It is possible that disarranged epidermal patterns displayed success in this study due to extensive standardization efforts. However, due to the difficulty in normalization, there are considerable limitations to its reliable use in clinical practice. Replacing this term with previously vetted and defined dermatopathology terminology may be useful. For instance, epidermal hyperplasia, epidermal atrophy, acanthosis, digitated, papillated, reticulated, basilar proliferation, or budding are histological terms used to describe changes to the epidermal architecture. Applying these terms to RCM may enable more specific descriptions that are readily understood and disseminated, as they have already been widely adopted and defined in histopathology Although some studies have reported significant reductions in nucleated cells [11,25,29] and dendritic cells [8,12] our results did not indicate a reliable, significant change in those parameters.

Actinic keratoses arise in areas of photodamaged skin and thus are evaluated in that setting. This has important implications in selecting which RCM parameters should be most heavily weighted for diagnosis and evaluation of AK. For instance, we observe that the baseline comparison of AK and PD skin (Table 3) yielded different key criteria than solely considering baseline percent prevalence alone (Appendix A). To efficiently distinguish AK from surrounding sun damaged areas, characteristics of the surrounding skin must be considered. In addition, PD skin can also serve as a reference point for the “normalization” of the RCM parameters. It is important to note that in field-directed interventions, the therapeutic effect on the PD skin can also be observed.

One challenge to RCM is the ability to image thick, hyperkeratotic lesions. As the thickness increases, the edge sensitivity and resolution decrease [30]. Some reports of RCM in AK exclude thicker lesions and most that examine a variety of AKs do not stratify their results based on grade, thus limiting their conclusions across lesion types. By studying and analyzing G1 and G2 AK separately, we found that the baseline odds ratios were higher in thicker (G2) AKs than thinner (G1) AKs, suggesting that pathologic RCM criteria are more easily detected in higher-grade AKs. Additionally, more parameters displayed a statistically significant decrease at 6 months in G1 AKs as compared to G2 AKs, demonstrating that obtaining a durable treatment response is more feasible in thinner AKs.

Previous reports have demonstrated that RCM allows for the increased detection of subclinical AK [8,11,24,25,28]. Ulrich et al. concluded that architectural disruption and cellular pleomorphism best characterize subclinical AK [5]. Ishioka et al. reinforced these findings, reporting stratum corneum disruption and atypical honeycomb patterns as the most prevalent subclinical AK characteristics post-treatment [11]. We observed the rebound of multiple criteria from 3 to 6 months, which likely indicated subclinical, early signs of relapse, particularly in G2 AKs. The criteria with the largest increases from 3 to 6 months included stratum corneum disruption, parakeratosis, and intraepidermal inflammatory cells. These findings suggest that, along with atypical honeycomb patterns, stratum corneum disruption, parakeratosis, and intraepidermal inflammatory cells may also serve as a clue for early AK recurrence. It is important to note that this evidence of early relapse was observed via RCM even though clinical evaluation demonstrated a sustained response. It has been suggested that all AK lesions, regardless of thickness, have a significant risk for invasive progression [31]. This emphasizes the importance of surveillance and supports the utility of RCM for the non-invasive monitoring of subclinical disease, and identification of early post-treatment AK recurrence.

Subject to operator-dependence and differences in individual experience, interobserver variability is an inherent limitation to RCM [32]. Similar studies assessing AK treatment response through RCM have frequently incorporated two experienced evaluators, and some have included a single experienced reader [11,12,13,24,25,26,28]. In our study, interobserver agreement was measured between three experienced RCM evaluators. We have shown that through focused consensus meetings and standardization efforts, interobserver agreement can be improved. This can be best exemplified through the parameters of stratum corneum disruption and disarranged epidermal pattern. Disruption of the stratum corneum may be difficult to assess in images obtained with a tangential angle, due to the anatomical location of the skin. Through consensus meetings, it was agreed that scattered evidence of black lagoons and a presence of isolated, hyperkeratotic scales favors the presence of SC disruption, while abrupt geometrical loss of the field favors artifacts, due to the tangential angle at the time of image acquisition. Disruption of the epidermal pattern was defined as disruption of the architectural arrangement of the honeycomb pattern. Readers agreed that it was important to evaluate the surrounding epidermis pattern before deciding on the presence or absence of this feature, as even normal skin may have a certain degree of disarrangement due to multiple factors, including tangential imaging or artifacts. Consensus significantly increased interrater agreement. A similar standardization approach may be useful in future trials with multiple evaluators. Additionally, we demonstrated that even if parameters only display moderate agreement, significant changes can still be observed. For instance, though stratum corneum disruption was among the lowest proportion of agreement (61%, CI: 57–65), there was still a significant decrease in the RCM criteria at both time points, in both treatment groups.

Clinical trials have demonstrated efficacy of both PDT and cryotherapy in the treatment of AK; however, PDT has demonstrated increased efficacy in head-to-head comparisons [33]. Although we did not identify a significant difference between treatment groups, both individually demonstrated success in the treatment of AK, as measured by both RCM and clinical evaluation.

Limitations of this study include the length of follow-up and the number of patients. A larger sample size would aid in comparing PDT and cryotherapy treatment. Additionally, the sample size of 20 participants affected the width of confidence intervals for the odds ratios that are listed in Table 3. Given that the clinical scores in Table 4 are allotted a half-value score from zero to three, the standard deviation of these scores had expanded due to the variability of these measurements. Furthermore, this study was also limited by the length of follow-up. Expanding the study length past 6 months would have better characterized the recurrence of AK. Trials published after our data were collected suggest that the classification of AK by the degree of basal proliferation and growth patterns is a more reliable indicator of progression to invasive carcinoma than grading based on the degree of cytological atypia [31,34,35,36]. Our definition of disarranged epidermal pattern encompasses the basal proliferation patterns described in these recent reports. Future studies should evaluate whether the incorporation of a more “granular PRO-like” AK scoring system, or a scoring system based on the growth pattern of basal keratinocytes, could also be applied to RCM. Additional consensus studies to further refine RCM terminology and descriptors, followed by larger studies’ assessments of a wider range of RCM parameters, are necessary to progress this initial work in creating a standardized, universal protocol for RCM monitoring of AK treatment responses. This step is critical if we are to establish a more objective, sensitive, and reliable methodology to comparatively assess the multiple AK treatment options and lead the field of therapeutic prevention into a cost-effective practice.

## 5. Conclusions

We identified RCM parameters that both effectively differentiated AK from nearby, photodamaged skin and reliably assessed AK responses to treatment. There is great value in the implementation of RCM in AK therapeutic response trials, as non-invasive, objective monitoring of lesions over time will lead to the more reliable characterization of longitudinal responses. A standardized, universally accepted protocol will decrease subjectivity, simplify clinical practice, and allow for the more meaningful comparison of results amongst clinical trials. Furthermore, the cellular level resolution obtained by RCM, unlike clinical scoring, provides insight into the biological mechanisms of treatment responses and thereby could identify actionable data towards optimizing therapy.

## Figures and Tables

**Figure 1 cancers-13-05488-f001:**
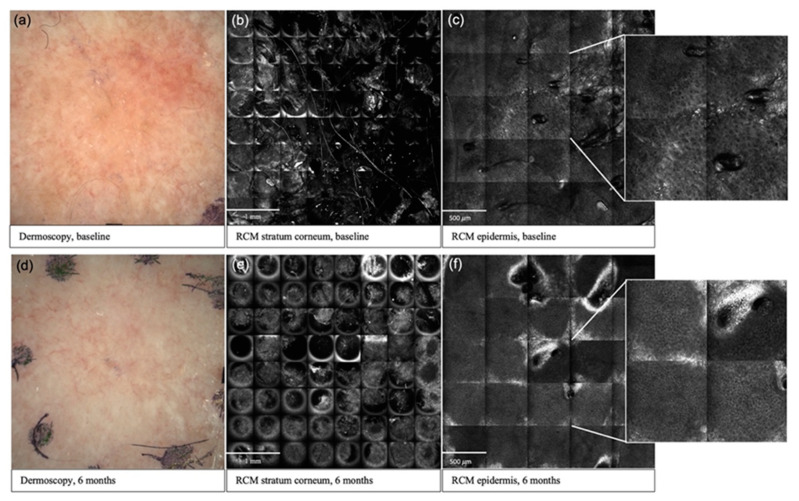
Clinical and reflectance confocal microscopy images over time in an AK lesion. Baseline (**a**) and 6 months (**d**) dermoscopic assessment of a G1 AK in the cryotherapy group; improvement in stratum corneum disruption from baseline (**b**) to 6 months (**e**) post-cryotherapy; improvement in atypical honeycomb pattern from baseline (**c**) to 6 months (**f**) post-cryotherapy.

**Table 1 cancers-13-05488-t001:** Clinical scoring based on measures of erythema and hyperkeratosis, with green indicating complete response at 6 months, orange indicating partial response at 6 months, and pink indicating clinical progression.

	Grade 1 AK						Grade 2 AK					
Subject ID	Baseline Erythema ^1^	3 Mo Erythema	6 Mo Erythema	Baseline Hyperkeratosis ^1^	3 Mo Hyperkeratosis	6 Mo Hyperkeratosis	Baseline Erythema ^1^	3 Mo Erythema	6 Mo Erythema	Baseline Hyperkeratosis ^1^	3 Mo Hyperkeratosis	6 Mo Hyperkeratosis
C1	1	0	0	1	0	0	2	0	0	2	0	0
C2	1	0	1	1	0	0.5	2	1	0	2	2	0.5
C3	1	0	0	1	0.5	0	2	0	0	2	0	0
C4	1	0	0	1	0	0	2	0	0	2	0	0
C5	1	0	0	1	0	0	2	0	0	2	0	0
C6	1	0	0	1	0	0	2	0	0	2	0	0
C7	1	0	0	1	1	0	2	0	0	2	0	0
C8	1	1	0	1	1	0	2	0	0	2	0	0
C9	1	1	0	1	0	0	2	1	0	2	0	0
C10	1	0	0.5	1	1	0	2	0	1	2	1	2.5
P1	1	0	0	1	0	0	2	0	0	2	0	0
P2	1	0	0	1	0	0	2	-	0	2	-	0
P3	1	0	1	1	0	0	2	0	0	2	1	0
P4	1	0	0	1	1	0.5	2	0	1	2	0	1
P5	1	0.5	0.5	1	0	0	2	0	0	2	0	0
P6	1	0	0	1	1	0	2	0	0	2	0	0.5
P7	1	-	-	1	-	-	2	-	-	2	-	-
P8	1	0	0	1	0.5	0	2	0	1	2	1	1
P9	1	0	0	1	0	0	2	0	0	2	0	0
P10	1	0	0	1	0.5	0	2	0	0	2	0.5	0.5

^1^ Per protocol, G1 AK had baseline clinical score of “1” for erythema and hyperkeratosis, and G2 AK had a baseline clinical score of “2” for erythema and hyperkeratosis. Clinical scoring was performed on a scale of 0 (absent) to 3 (3+). Subject P7 did not receive clinical scoring.

**Table 2 cancers-13-05488-t002:** Interobserver agreement of RCM parameters for AK.

Skin Layer	RCM Parameter	Pre-Standardization Proportion of Agreement ^1^ (95% CI)	Post-Standardization Proportion of Agreement (95% CI)
Stratum Corneum	Parakeratosis	54 (45–63)	65 (60–69)
	Hyperkeratosis	63 (53–71)	63 (58–68)
	Stratum Corneum Disruption	37 (24–52)	61 (57–65) ^SS^
Epidermis	Atypical Honeycomb Pattern ^2^	80 (68–88)	86 (83–88)
	Round Nucleated Cells	52 (42–61)	79 (76–82) ^SS^
	Disarranged Epidermal Pattern	37 (29–47)	63 (59–66) ^SS^
	Presence of Inflammatory Cells	61 (55–67)	62 (58–66)
	Presence of Dendritic Cells	87 (80–92)	80 (77–83)
Dermis	Presence of Inflammatory Cells ^2^	85 (82–89)	83 (80–86)
	Solar Elastosis	N/A	73 (69–76)
	Round Blood Vessels ^2^	69 (57–78)	64 (60–67)
	Polymorphous Blood Vessels ^2^	53 (74–65)	81 (78–84) ^SS^

^1^ Pre-standardization time-point: prior to any RCM consensus meetings or standardization (pre-study). Pre-standardization readers included CCL, RHW, and GP. Readers for post-standardization proportion of agreement, following subject RCM analysis, included GP, CCL/EA*, MU (cryotherapy) and GP, CCL/EA*, MO (PDT). *CCL/EA read images together. ^2^ Non-dichotomous variables, agreement within 1 point. ^SS^ Indicates a statistically significant improvement in interobserver agreement.

**Table 3 cancers-13-05488-t003:** Odds ratios comparing RCM parameters in AK to PD skin at baseline.

Skin Layer	RCM Parameter	Grade 1 AK (95% CI)	Grade 2 AK (95% CI)	All AK (95% CI)
Stratum Corneum	Parakeratosis	5.5 ** (2.6–11.7)	5.3 ** (2.3–11.9)	5.3 ** (2.6–10.6)
	Hyperkeratosis	10.6 ** (3.8–29.0)	18.3 ** (6.1–54.5)	13.6 ** (5.3–34.9)
	Stratum Corneum Disruption	7.1 ** (2.9–17.4)	8.9 ** (3.5–22.8)	7.8 ** (3.5–17.3)
Epidermis	Atypical Honeycomb Pattern ^1^	9.7 ** (3.8–24.5)	18.0 ** (7.7–41.8)	12.7 ** (5.7–28.1)
	Round Nucleated Cells	3.0 * (1.01–8.9)	1.6 (0.5–5.1)	2.3 (0.8–6.5)
	Disarranged Epidermal Pattern	5.4 ** (2.2–12.3)	8.3 ** (3.2–21.5)	6.5 ** (2.9–14.8)
	Presence of Inflammatory Epidermal Cells	1.9 (0.9–4.1)	2.4 * (1.1–5.2)	2.1 * (1.1–4.0)
	Presence of Dendritic Cells	0.8 (0.2–2.6)	1.1 (0.3–3.5)	0.9 (0.4–2.6)
Dermis	Inflammatory Infiltrate Dermis ^1^	3.6 ** (1.5–8.6)	2.2 (0.8–5.8)	2.8 * (1.2–6.5)
	Solar Elastosis	3.5 (0.9–12.5)	1.6 (0.6–4.4)	2.2 * (0.9–5.5)
	Round Blood Vessels ^1^	1.5 (0.8–2.7)	1.8 * (1.02–3.2)	1.6 * (1.0–2.6)
	Polymorphous Blood Vessels ^1^	3.8 ** (1.6–8.7)	3.0 * (1.1–7.7)	3.3 ** (1.5–7.4)

Scores from RCM evaluators are averaged to obtain odds ratios. Odds ratios are presented for AK lesions compared to PD skin, * *p* < 0.05, ** *p* < 0.01. ^1^ For non-dichotomous variables, odds ratios were calculated using ordinal logistical regression.

**Table 4 cancers-13-05488-t004:** Mean (standard deviation) clinical scores over time and RCM parameter prevalence over time in cryotherapy and PDT groups.

		Treatment	Grade 1 AK			Grade 2 AK			PD Skin		
			BL	3 mos	6 mos	BL	3 mos	6 mos	BL	3 mos	6 mos
Clinical Parameter	Erythema	Cryo	1.00 ^1^	0.20 (0.42) **	0.15 (0.34) **	2.00 ^1^	0.20 (0.42) **	0.10 (0.32) **	N/A	N/A	N/A
		PDT	1.00 ^1^	0.55 (0.17) **	0.17 (0.35) **	2.00 ^1^	0.00 (0.00) **	0.22 (0.44) ** ^2^	N/A	N/A	N/A
	Hyperkeratosis	Cryo	1.00 ^1^	0.35 (0.47) **	0.05 (0.16) **	2.00 ^1^	0.30 (0.67) **	0.30 (0.79) **	N/A	N/A	N/A
		PDT	1.00 ^1^	0.33 (0.43) **	0.05 (0.17) **	2.00 ^1^	0.31 (0.46) **	0.33 (0.43) ** ^3^	N/A	N/A	N/A
RCM Parameter: Stratum Corneum	Parakeratosis	Cryo	85%	43% **	57% *	58%	50%	70%	22%	31%	37%
		PDT	52%	29%	29%	79%	16% **	50%	33%	12%	44%
	Hyperkeratosis	Cryo	71%	45% *	23% **	80%	47% **	43% **	19%	21%	17%
		PDT	45%	22%	20%	63%	33% *	56%	5%	6%	11%
	Stratum Corneum Disruption	Cryo	88%	47% **	40% **	83%	60% *	67%	26%	46%	40%
		PDT	56%	39%	12% *	69%	24% *	47%	26%	22%	13%
Epidermis	Atypical Honeycomb Pattern ^4^	Cryo	100%	67% *	63% **	96%	83% *	83% **	65%	57% *	57%
		PDT	97%	63% *	53% **	100%	69% **	78% **	73%	59%	59%
	Round Nucleated Cells	Cryo	23%	23%	7%	13%	7%	7%	4%	7%	7%
		PDT	23%	4%	6%	15%	19%	33%	14%	0%	6%
	Disarranged Pattern	Cryo	48%	23% *	13% **	59%	38% *	13% **	11%	7%	17%
		PDT	53%	16% *	25%	62%	35% *	44%	20%	11%	12%
	Inflammatory Cells Present	Cryo	61%	40%	28% *	43%	24%	53%	35%	27%	30%
		PDT	50%	22% *	22%	74%	35% **	41% *	43%	18% *	29%
	Dendritic Cells Present	Cryo	18%	10%	17%	13%	10%	23%	11%	10%	23%
		PDT	4%	8%	0%	15%	4%	6%	14%	0%	12%
Dermis	Inflammatory Cells Present ^4^	Cryo	61%	54%	27% *	39%	29%	40%	31%	40%	37%
		PDT	65%	48% **	29% *	54%	50%	21%	34%	37%	29%
	Solar Elastosis	Cryo	96%	79%	90%	81%	68%	87%	70%	83%	63%
		PDT	89%	96%	100%	88%	93%	100%	86%	92%	88%
	Round Blood Vessels ^4^	Cryo	67%	79%	77%	64%	72%	73%	61%	70%	73%
		PDT	72%	85% *	94% *	86%	81%	100%	62%	84% **	94% **
	Polymorphous Blood Vessels ^4^	Cryo	44%	31%	13%	26%	21%	17%	19%	13%	13%
		PDT	38%	23%	23%	43%	38%	29%	14%	32% *	23%

Significance compared to baseline (BL), * *p* < 0.05, ** *p* < 0.01. ^1^ Per protocol, G1 AK had baseline clinical score of “1” for erythema and hyperkeratosis, and G2 AK had a baseline clinical score of “2” for erythema and hyperkeratosis. Clinical scoring was performed on a scale of 0 (absent) to 3 (3+). ^2^
*p*-value from 3 to 6 months = 0.6815. ^3^
*p*-value from 3 to 6 months = 0.8702. ^4^ For non-dichotomous variables, percent prevalence calculated using ordinal logistical regression.

## Data Availability

The data presented in this study are available on request from the corresponding author.

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
