# Peer review of "In Vivo Reflectance Confocal Microscopy as a Response Monitoring Tool for Actinic Keratoses Undergoing Cryotherapy and Photodynamic Therapy"

_cancers, 2021, doi:10.3390/cancers13215488_

Round 1

Reviewer 1 Report

Despite its significant limitations (number of patients, follow up period, limited popularity of RCM among dermatologists), the article makes a valuable contribution to the management of actinic keratoses. It is important to look for reliable and objective criteria for AK's response to treatment. The article is well written and has some merits, the limits of this study are presented transparently. Still, there are some concerns about the work.

  • (106-108, Table 1) The first concern relates to clinical classification. The authors use two criteria: 1) erythema (using dermatoscopy), 2) hyperkeratosis (according to Olsen). The degree of erythema and the degree of hyperkeratosis (e.g. 1 = mild, 2 = moderate and 3 = severe) must be specified here. Please explain the meaning of 1+, 2+, 3+ to readers as Cancers has a broad readership outside of dermatology. In this context, how should the reader understand the meaning of the unnatural numbers given in Table 1? Example: Subject C10 after 6 months: Erythema = 0.5. Absent or present? Since the evaluation was made by a single investigator, it is also not the average. This could affect the statistics of the clinical evaluation and should be improved prior to publication.
  • The authors should discuss in detail the reason for the very large confidence intervals for most of the odd ratios of the RCM parameters (Table 3), including hyperkeratosis and atypical honeycomb pattern. The large standard deviations of the clinical scores as indicated in Table 4 should also be discussed. Please also standardize the layout of tables 3 and 4 (e.g. for p <0.01 use the same special character in both tables, i.e. replace ¥ with **).
  • (289-291) It might be helpful to explain in detail what was the main reason for the substantial improvement in the intraobserver agreement of RCM parameters such as stratum corneum disruption or disarranged epidermal pattern. Is it possible beyond the short sentence (349-350) to name criteria for the normalization of these parameters? Perhaps RCM images would help the reader understand the problem better. Please also state what exactly is meant by "Replacing this term with previously vetted and defined dermatopathology terminology may be useful".
  • (352) "Granular PRO-like": For a better understanding, please add a brief explanation that the PRO classification of AKs is based on the growth pattern of basal keratinocytes.

Minor issues:

  • (140) J/cm2
  • In Abstract atypical honeycomb CI 5.7-21.5, but in Table3: 5.7-28.1

Author Response

Please see the attachment, with a point-by-point response to the reviewer's comments added as a cover letter. 

Reviewer 2 Report

In this manuscript, the authors have identified reflectance confocal microscopy parameters that efficiently differentiate actinic keratosis (AKs) from neighboring photodamaged skin and also at the same time reliably assess AKs response to treatment. This is a well-written manuscript in which the authors have nicely compiled the findings in the form of tables and figures. However, the authors should consider discussing and citing some of the recent publications in the areas of cryotherapy and photodynamic therapy.

The authors aimed to identify reliable actinic keratoses (AKs) response criteria and compared clinical and reflectance confocal microscopy (RCM) evaluation response across AK severity grade. For these studies, twenty patients were included and randomized to receive either cryotherapy or photodynamic therapy (PDT). The presented data indicate that both treatments decreased AK severity as measured by RCM. In addition, both the PDT and cryotherapy treatments were tolerated by all patients. The findings of this study also suggest that along with atypical honeycomb pattern, stratum corneum disruption, parakeratosis, and intra-epidermal inflammatory cells may serve as a clue for early AK recurrence. Furthermore, the authors identified RCM parameters that both effectively differentiate AK from nearby photodamaged skin and reliably assess AK response to treatment. The manuscript is well-written and presented. However there are some concerns which are provided below.

  1. The sample size (20 patients) and the duration of this study (3 to 6 months) are the major limitations. The authors should consider addressing these issues in the revised manuscript.
  2. The authors should discuss in this manuscript how their study is different for the earlier published research articles on AK (Dermatology 2010, 220, 15–24; J Eur 422 Acad Dermatol Venereol 2018, 32, 1155–1163).
  3. The authors should also consider discussing and citing some of the recent publications in the area of photodynamic therapy (PMIDs: 34633154, 33482254).

Round 2

Reviewer 1 Report

The authors have addressed all queries appropriately. The manuscript has been improved and I have no more concerns about the manuscript. Congratulations to the authors.